Searching for a common host: parasitoids of Lema daturaphila on Datura stramonium in Central Mexico

Villanueva-Hernández Carol Estefanía
http://orcid.org/0000-0001-5829-8338 Núñez-Farfán Juan farfan@unam.mx
Department of Evolutionary Ecology, Institute of Ecology, Universidad Nacional Autónoma de México , Ciudad de México , Mexico
Sunny Armando
Electronic publication date: 2025 Feb 3
Publication date: 2025
Volume: 13
Electronic Location ID: e18675
Received 2023 Mar 21; Accepted 2024 Nov 19
Copyright: © 2025 Villanueva-Hernández and Núñez-Farfán
Copyright year: 2025
Copyright holder: Villanueva-Hernández and Núñez-Farfán
License: This is an open access article distributed under the terms of the Creative Commons Attribution License, which permits unrestricted use, distribution, reproduction and adaptation in any medium and for any purpose provided that it is properly attributed. For attribution, the original author(s), title, publication source (PeerJ) and either DOI or URL of the article must be cited.
License URL: https://creativecommons.org/licenses/by/4.0/

Keywords: Tritrophic interactions, Plant herbivore-interactions, Plant defense, Oral secretions, Tachinidae, Eulophidae, Parasitoid infestation, Chrysomelidae, Egg parasitoids, Egg clutch

Funding: DGAPA PAPIIT UNAM IN226823 Sistema Nacional de Investigadores (SNI, CONACYT) CONAHCyT granted for Doctoral Studies at the Posgrado en Ciencias Biológicas at UNAM This work was supported by DGAPA PAPIIT UNAM IN226823. Carol Estefanía Villanueva-Hernández received a scholarship from Sistema Nacional de Investigadores (SNI, CONACYT), and from CONAHCyT granted for Doctoral Studies at the Posgrado en Ciencias Biológicas at UNAM. The funders had no role in study design, data collection and analysis, decision to publish, or preparation of the manuscript.

==============================
Background

Natural enemies of herbivore insects can change the arms race between plants and insects. Their presence and abundance even can affect the co-evolution of interacting species. The annual herb Datura stramonium varies geographically in the extent of its direct defenses against herbivores. Its main specialist herbivore, Lema daturaphila, is adapted to cope with these defenses, but little is known about its natural enemies. Here, we determined the presence and incidence of L. daturaphila parasitoids as an initial step to explore other ecological and evolutionary relationships in a tri-trophic context.

Methods

Field collections of L. daturaphila eggs and larvae were performed during the summers of 2018 and 2019 in eleven natural populations of D. stramonium in central Mexico. We recorded their development to evaluate the emergence of parasitoids and their relationship with the abundance of herbivore individuals and environmental variables in each locality.

Results

We found six parasitoid fly and wasp species that are new records for Mexico or the host. Throughout their distribution, the interaction varies widely among populations and years. In some localities, egg parasitoids dominate over larval parasitoids and vice versa, and they exert strong pressures on the survival of L. daturaphila’s populations. The abundance of Emersonella lemae, the egg parasitoid, is related to the clutch size of L. daturaphila and climatic conditions such as temperature, altitude, and precipitation. As an apparent defense strategy against parasitoid flies, larvae of L. daturaphila release their oral secretions, which contain alkaloids from D. stramonium. At a geographic scale, these findings change the scenario between the plant-herbivore interaction and open the field to explore the different selective pressures among populations.

Introduction

Plants and herbivorous insects maintain antagonistic relationships mediated by the coevolution of their defense-counter defense traits (Ehrlich & Raven, 1964; Futuyma & Agrawal, 2009). Plants have evolved physical defensive mechanisms like trichomes and spines, or specialized chemical metabolites to defend themselves against insects (Hilker & Fatouros, 2015; Bittner, Trauer-Kizilelma & Hilker, 2017; Turlings & Erb, 2018). In response, insects can withstand, counterattack, and even make plants more vulnerable to subsequent damage, promoting and giving rise to arms races (Karban & Agrawal, 2002; Chung et al., 2013). Besides these direct interactions, plants can deal with insects through indirect defenses which involve, among others, the recruitment of natural enemies from the third trophic level (Turlings & Erb, 2018). These enemies reduce herbivore pressure, affect and are affected by plant traits and by the herbivore quality, contributing to the selective pressures imposed between plants and herbivores (Price et al., 1980; Ode, 2006).

The complexity of tritrophic interactions extends beyond defense mechanisms and involves geographic factors and local environmental conditions, that shape coevolutionary dynamics (Craig, Itami & Horner, 2007). Plants and their associated insects are distributed along wide geographic areas, where the physical environment and the community structure of different interacting species could generate differential local pressures (Craig, Itami & Horner, 2007; Connahs et al., 2009). These differences contribute to the formation of a Geographic Mosaic of Coevolution, with plants, herbivores, and natural enemies interacting differentially and contrastingly under variable local conditions (Althoff & Thompson, 1999; Thompson, 2005).

Parasitoids are natural enemies that cause high mortality in their host insects (Hawkins, Cornell & Hochberg, 1997; Ode, 2006). They can significantly influence plant-herbivore dynamics (Ode, 2006). Plants’ volatile compounds can attract them and are used as cues to find their host (Birkett et al., 2003; Ngumbi, Chen & Fadamiro, 2009; Morawo & Fadamiro, 2016). Since parasitoids reduce herbivores’ pressure on plants, they are generally considered plant mutualists (Takabayashi et al., 1998; de Lange et al., 2018). However, they can also be affected by plant chemistry, whether by direct exposition to these substances or indirectly by contact with hosts (Wajnberg & Colazza, 2013). This is particularly important when the bridge between plant and parasitoid is a host-specialized herbivore. Herbivores with trophic specialization might use plant chemical defenses to their advantage, as a chemical weapon against parasitoids (Bernays & Graham, 1988; Gross, 1993; Ali & Agrawal, 2012). For instance, oral secretions (OS) are blends composed of pieces of plants, saliva, digestive enzymes, and toxic plant metabolites. These OS are released by herbivores when attacked by natural enemies (Sword, 2001), thus constituting an effective way to exploit plant toxicity and reduce the likelihood of parasitoidism (Barbosa et al., 1986; Gross, 1993; Karban & Agrawal, 2002).

An ideal system to explore these kinds of ecological and evolutionary relationships is the interaction between Datura stramonium (Solanaceae), its specialist herbivore Lema daturaphila (Chrysomelidae), and its associated parasitoids. In Central Mexico, the direct defenses of D. stramonium, such as leaf trichomes and tropane alkaloids, vary geographically according to the dominant herbivore guild in each locality (Castillo et al., 2014). Scopolamine, the most toxic alkaloid in D. stramonium, appears to be more effective against generalists than specialist herbivores (Castillo et al., 2013, 2014). Among the latter, L. daturaphila is the most harmful insect for D. stramonium (Núñez-Farfán & Dirzo, 1994). In some populations, this beetle has shown to be adapted to overcome the defenses of its host plant (Núñez-Farfán & Dirzo, 1994; Castillo et al., 2013; De-la-Cruz et al., 2020).

Lema daturaphila can tolerate high amounts of atropine and scopolamine, the main alkaloids in Datura (Kogan & Goeden, 1971; Weaver & Warwick, 1984; Shonle & Bergelson, 2000). It even can reduce their production, apparently suppressing these chemical defenses in the plant (Zhang et al., 2022). The larvae of L. daturaphila release oral secretions when disturbed, which may serve as a defense against natural enemies (Omer-Cooper & Miles, 1951; Cabrales-Vargas, 1991). These findings highlight the strong interaction between L. daturaphila and its host plant and suggest that parasitoids may play a crucial role as indirect defenses of the plant, reducing fitness costs to the plants by L. daturaphila.

However, there is still a gap in the knowledge of L. daturaphila’s parasitoids and the potential defenses it uses against them (Cabrales-Vargas, 1991; Garrido, 2004; Hernández-Cumplido, 2006). Since L. daturaphila exerts strong selection pressure on D. stramonium (De-la-Cruz et al., 2020), exhibiting resistance to the plant’s direct defenses in several localities, we hypothesize that parasitoids may be key regulators of L. daturaphila populations and serve as an indirect defense for D. stramonium against this specialist herbivore. We hypothesize that interpopulation variability in the plant-herbivore interaction is associated with the presence of parasitoids and that certain populations can be considered coevolutionary hotspots (Thompson, 1999), where the tritrophic interaction is stronger. Likewise, we hypothesize that oral secretions could be a defense strategy of L. daturaphila, which uses the chemical composition of its host plant to cope with parasitoids. In this study, we analyzed the tritrophic interaction between Datura stramonium-Lema daturaphila and its parasitoids across natural populations in Central Mexico. Our goal was to examine this interaction throughout different geographic regions to test the hypothesis outlined before.

Materials and Methods

Study system

In Mexico, Datura stramonium is commonly found in human-disturbed areas (Núñez-Farfán & Dirzo, 1994), where it is consumed by folivore insects such as Epitrix parvula and Sphenarium purpurascens, as well as by the pre-dispersal seed predator Trichobaris soror (Castillo et al., 2015; De-la-Mora, Piñero & Núñez-Farfán, 2015). However, Lema daturaphila stands out because its gregarious larvae can cause defoliation as high as 100% in some D. stramonium populations (Núñez-Farfán & Dirzo, 1994). Except for the pupa, all the stages of its life cycle occur on the plant, beginning when females oviposit clutches of 15–30 ovoid eggs on the underside of the leaves (Kogan & Goeden, 1970). The bright yellow eggs, which are around 1 mm long, gradually darken until hatching (Omer-Cooper & Miles, 1951). Lema daturaphila larvae then passes through four instars, which are morphologically similar to each other (Kogan & Goeden, 1970). The larvae deposit a mass of feces on their backs, and their oral secretions are dark brown fluids that larvae can suck back (Omer-Cooper & Miles, 1951). In these immature stages, L. daturaphila larvae can be target of unidentified tachinid flies and hymenopteran parasitoids (Omer-Cooper & Miles, 1951; Cabrales-Vargas, 1991; Hernández-Cumplido, 2006). Additionally, an ancient record mentions Emersonella lemae, a chalcid wasp that attacks L. daturaphila eggs on D. stramonium in Washington, D.C. (Girault, 1916; Chittenden, 1924). When the last instar is ready, the larvae leave the plant and build a cocoon in the soil where they pupate until the adult emerges and returns to the plant to feed and reproduce (Kogan & Goeden, 1970).

Sampling

Between August and October 2018 and 2019, we sampled nine natural populations of D. stramonium and L. daturaphila in Central Mexico (Fig. 1). Their geographic location and climatic conditions are detailed in Table S1. The collection of L. daturaphila was authorized by the Secretaría de Medio Ambiente y Recursos Naturales, Gobierno de México (Oficio N° GPA/DGVS/011980/17). In the second year of sampling, we included the populations of Dolores and San Martín, located in the states of Querétaro and Puebla, respectively.

Figure 1 Sampled populations of Datura stramonium and Lema daturaphila in central Mexico.

Details of each locality are described in Table S1. The shape used to representing the sampling localities was taken from the CONABIO geoportal: http://www.conabio.gob.mx/informacion/gis/ with data of (INEGI, 2024) and (CONABIO, 2024). License: CC BY-NC 2.5 MX, https://creativecommons.org/licenses/by-nc/2.5/mx/.

In each population, we randomly collected ±30 L. daturaphila egg clutches and, additionally, ±30 individuals of the 2° or 3° larval instars. All individuals were reared under controlled conditions (22 °C, 12:12 h photoperiod). We recorded the number of eggs per clutch and individual clutches were separated into Petri dishes. Larvae were transferred to plastic containers and fed with leaves from a natural D. stramonium population in the Pedregal de San Angel natural reserve (southern Mexico City). All eggs and larvae were inspected daily for emerging chrysomelids or parasitoids. Emerging parasitoids were preserved in 70% ethanol. They were examined with a LEICA MZ125 binocular microscope and determined using taxonomic keys with assistance from entomologist specialists (Wood, 1987; Hansson, 2002; Fernández & Sharkey, 2006). Representative specimens were photographed using a Nikon D3400 digital camera and with a Hitachi SU1510 scanning electron microscope. Photographs were processed with ON1 Photo Raw 2022 and Inkscape 1.3.

Analysis of oral secretions

We selected non-parasitized egg clutches from Bernal, Querétaro, and reared L. daturaphila larvae under the same environmental conditions mentioned above. Oral secretions were collected by squeezing a larva behind its head, inducing immediate regurgitation. Using a micropipette, we collected 1–5 µL of oral secretion per larva. The crude oral secretions from 10–15 larvae were stored in Eppendorf tubes at −20 °C until further processing.

The chemical composition of the oral secretions was analyzed using a liquid chromatography/time-of-flight/mass spectra (HPLC-TOF-MS) system, following the protocol described for D. stramonium by De-la-Cruz et al. (2020). First, the crude oral secretions were centrifuged at 12,000 rpm for 30 min to remove pieces of plant material and other solids. The supernatant was then centrifuged under the same conditions and filtered through a 0.22 µm sterilized membrane.

The sample was re-suspended in 100 µL of MeOH and centrifuged at 12,000 rpm for 2 min. The supernatant was recovered, re-suspended in 500 µL of MeOH, and centrifuged again. Finally, chromatographic separations were performed on an Agilent ZORBAX HPLC column, following the De-la-Cruz et al. (2020) protocol. Chemical compounds were identified using mass spectrometry, retention times, and molecular formulas obtained from chromatograms with MassHunter Workstation software (version B. 06.00; Agilent Technologies, Santa Clara, CA, USA).

Data analysis

All statistical analyses and graphs were performed using R version 4.3.1 (R Core Team, 2021) and the ggplot2 package (Wickham, 2016).

We first conducted descriptive statistical analysis to examine the variation in the number of eggs and larvae of L. daturaphila, as well as parasitoid individuals. Given that the data were counts and there was high variance in the number of eggs per clutch, we used the MASS package (Venables & Ripley, 2002) to fit a generalized linear model (GLM) with a negative binomial distribution. This model allowed us to assess differences in clutch size across populations, years, and their interaction. Next, to assess the incidence of parasitized egg clutches (binomial response: parasitized or non-parasitized), we performed a GLM with a binomial distribution, including the interaction between population and year.

To determine whether the number of wasps emerging per clutch varies across populations and years, we fit a GLM with a negative binomial distribution, where the number of eggs per clutch and population were used as predictor variables. We also performed a logistic regression to evaluate whether the probability of parasitism depends on clutch size each year. For modeling parasitoidism in L. daturaphila larvae, we used a GLM with a Poisson distribution. Since we only collected a sample of larvae in each locality, the number of larvae per population was the sole predictor variable. Finally, to explore patterns in parasitoid incidence and their relationship with environmental variables, we performed a principal component analysis (PCA), including average climatic conditions per year from each locality.

Results

During the first sampling period (August to October 2018), we collected 195 egg clutches and 197 L. daturaphila larvae. The following year (August to September 2019), we collected 411 egg clutches and 838 L. daturaphila larvae (Table S2). Six solitary endoparasitoid species from the orders Diptera and Hymenoptera were identified. We provide a brief review of each species with observations of their natural history.

Parasitoid species

The egg stage of L. daturaphila was parasitized by Emersonella lemae, an idiobiont wasp approximately 1 mm in length. This species belongs to the Eulophidae family (superfamily Chalcidoidea) and exhibits sexual dimorphism in the metasoma: females are round, while males have a more hexagonal shape (Fig. 2). Parasitism occurs shortly after L. daturaphila deposits its eggs (Fig. 3A). Multiple E. lemae females likely parasitize a single clutch, as we observed several wasps visiting the same clutch in the field. During L. daturaphila larval development, the eggshell remains transparent (Fig. 3B), but it turns opaque and golden-brown if parasitized (Fig. 3C). Healthy eggs hatch 5–7 days after oviposition, while wasp development takes over a month. Upon emergence, the adult wasp punctures the egg with its mouthparts, typically at the top of the egg (now a pupa) (Fig. 3D).

Figure 2 Emersonella lemae, egg parasitoid of Lema daturaphila.

Female in a (A) dorsal and (B) lateral view. Sexual dimorphism in the metasoma from female (C) and male (D). Lower photographs were taken with a scanning electron microscope. The above images were provided by Christer Hansson.

Figure 3 Parasitoidism on Lema daturaphila eggs by Emersonella lemae.

(A) Female of Emersonella lemae ovipositing on the chrysomelid eggs. (B) Healthy hatched eggs of L. daturaphila. (C) The semblance of a parasitized egg clutch. (D) Emersonella lemae wasps emerging from eggs.

In the larval stage of L. daturaphila, we identified four koinobiont flies from the genera Patelloa, Winthemia, Heliodorus, and Pseudochaeta. These parasitoids emerge during the pupal stage and are often found around larvae (Fig. 4A), which respond by releasing a bubble of oral secretions (Fig. 4B). After parasitization, the larvae continue feeding until pupation, at which point flies make a reddish-brown cocoon that hatches 15–20 days later (Figs. 4C, 4D). Only in the Pedregal population, we found a parasitoid wasp from the Ichneumonidae family parasitizing L. daturaphila larvae (Fig. S1A). In the Toluca and Tlaxiaca populations, Mesochorus sp., a hyperparasitoid wasp, emerged from fly cocoons (Figs. S1B, S1C).

Figure 4 Parasitoidism on Lema daturaphila larvae by Tachinidae flies.

(A) Parasitoid fly surrounding Lema daturaphila larva. (B) Larva excreting its oral secretion. (C) Fly pupae inside the dead body of Lema daturaphila, and (D) inside a cocoon from Lema daturaphila.

Population variability in parasitoid infestation

We observed high and significant variability in the abundance of L. daturaphila egg and larval stages and their parasitoids across populations. At least one parasitoid was associated with each population.

Clutch size significantly varied across populations and years (GLM, p < 0.05, Fig. 5, Table S3). Tlaxiaca and Tzintzuntzan had the highest mean clutch sizes in 2018 and 2019, respectively (23.78 ± 12.03; 33.93 ± 9.35, Table S4). In 2018, 46.15% of the total egg clutches collected were parasitized, and in 2019, this increased to 51.58% (Fig. 6). Requena, Texcoco, Tzintzuntzan and Valsequillo had a significantly higher parasitism (Table S5). In the first year, Tzintzuntzan, Valsequillo, and Requena exhibited the highest parasitism rates (91.6%, 84.3%, and 83.3%, respectively). However, no parasitism was observed in Teotihuacan and Toluca (Fig. 6). In 2019, parasitism of egg clutches was total in the Pedregal, Tzintzuntzan, and Valsequillo populations, while Bernal and Teotihuacan were uninfested. In contrast to 2018, Toluca had 8.5% parasitism of egg clutches (Fig. 6).

Figure 5 Among-populations variation in the number of eggs per clutch in 11 populations of Lema daturaphila in Central Mexico during 2018 and 2019.

The horizontal lines within each box denote median values, asterisks (*) represent statistically significant differences between both years in the marked populations (GLM, p < 0.05).

Figure 6 Parasitism of Lema daturaphila egg clutches per population in both years.

Bars indicate the number of egg clutches that were either parasitized or non-parasitized by Emersonella lemae, along with their corresponding percentage value. Asterisks (*) represent populations with a statistically significant higher level of parasitized egg clutches (GLM, p < 0.01).

Despite the high coverage of L. daturaphila egg clutches by E. lemae, not all eggs in a clutch were killed by parasitism; some completed their development or died from other causes. Egg mortality rates varied among populations. In 2018, Tzintzuntzan, Requena, and Valsequillo exhibited the highest mortality rates (67.25%, 61.67%, and 49.45%, respectively) (Fig. 7). Then in 2019, Pedregal and Requena had the highest egg mortality levels (96.8% and 93.03%, respectively). In Valsequillo, although E. lemae visited all clutches, only 49.45% of the eggs died from parasitism. Dolores and San Martín, the new censused populations for that year, had total parasitoidism rates of 34.42% and 22.01%, respectively (Fig. 7). In 2019, larger clutches of L. daturaphila were significantly preferred by E. lemae (GLM, p = 0.0254; Fig. 8, Table S6). Additionally, there was a significant positive relationship between the number of Lema´s eggs and the emerged parasitoid wasps per clutch in both years (Fig. 9, Table S7).

Figure 7 Population variation in parasitized eggs of Lema daturaphila by Emersonella lemae each year.

Bars indicate the percentage of parasitized and non-parasitized eggs per population. Asterisks (*) indicate populations where there was a significant increment in the number of parasitized eggs (GLM, p < 0.01). During 2019 all populations were statistically similar.

Figure 8 Parasitoidism by Emersonella lemae depends on the clutch size of Lema daturaphila in 2019.

Only in 2019 were the larger egg clutches of Lema daturaphila more parasitized (GLM, p = 0.0254). Each point represents an egg clutch from a corresponding population, their positions were jittered for better visualization. The regression line and confidence intervals were calculated using a binomial generalized linear model.

Figure 9 Relationship between emerged Emersonella lemae wasps and the clutch size of Lema daturaphila in both years.

Each point represents a Lema daturaphila egg clutch, estimates were calculated with a negative binomial generalized linear model and they were back-transformed to the original measure scale. In both years, the number of eggs per clutch had a significant effect on the number of wasps that emerged. However, just in 2018, Requena, Texcoco, Tzintzuntzán, and Valsequillo had a significant increase in the number of Emersonella lemae wasps compared with the reference population, Bernal (GLM, p < 0.05).

We detected Tachinidae flies in all populations of L. daturaphila except Pedregal. In some localities, the abundance of larvae was very low, but parasitism still occurred. Larvae were absent in the Tzintzuntzan population in 2018. The highest rates of larval parasitism were observed in Tlaxiaca, Texcoco, and Teotihuacan in both years (Fig. 10). The number of parasitoid flies showed a significant relationship with the number of larvae collected per population (GLM, p < 0.0001, Fig. 11, Table S8).

Figure 10 Variation in the percentage of Lema daturaphila larvae parasitized per population in both years.

Bars indicate the number of larvae that were parasitized and non-parasitized by Tachinid flies each year. Non-parasitized individuals successfully completed their development or died from other causes.

Figure 11 Relationship between emerged parasitoid flies and the number of larvae of Lema daturaphila in both years.

Each point represents a sample of Lema daturaphila larvae per population, estimates were calculated with a Poisson generalized linear model and they were back-transformed to the original measure scale. In both years, there was a significant effect of the number of larvae on the flies that emerged (GLM, p < 0.0001).

Analysis of environmental variables revealed divergent climatic conditions that influenced parasitoid presence and L. daturaphila populations. The first three principal components explained most of the variance (36.0%, 23.9%, and 20.76% respectively) (Table 1). Populations such as Pedregal, Tzintzuntzan, Valsequillo, and Requena, characterized by higher temperatures and precipitation, tended to have more parasitized egg clutches. Non-parasitized clutches were typically found at higher altitudes in populations such as Teotihuacan, Tlaxiaca, and Toluca. The number of emerged parasitoid wasps was negatively associated with the emergence of larvae in populations at higher altitudes and lower precipitation (Fig. 12).

Table 1 Values for the first three principal components for the variance in the incidence of parasitism and the environmental characteristics through populations.

	PC1	PC2	PC3	
Standard deviation	1.469	1.198	1.115	
Cumulative % of variance	36.0	59.94	80.70	
Eigenvectors				
Eggs per clutch	0.106	−0.370	0.763	
Lema daturaphila larvae	−0.434	0.188	0.553	
Emersonella lemae wasps	0.473	−0.488	0.114	
Altitude	−0.560	−0.358	0.097	
Precipitation	−0.011	−0.616	−0.281	
Temperature	0.511	0.283	0.091	
Note:

Variables with the highest charge on each principal component are shown in bold.

Figure 12 Principal component analysis of the variance in the incidence of parasitism and the environmental characteristics of each population.

PC1 and PC2 explain 59.94% of the variance. Variables included were the total number of eggs per clutch with the larvae and parasitoid wasps that emerged from those clutches. We included as environmental variables the temperature, altitude, and precipitation. The first three principal components explain 80.70% of the variation.

Analysis of Lema daturaphila oral secretions

Four tropane alkaloids were detected in the oral secretions of larvae of L. daturaphila (Table 2). These compounds are also present in its host plant, D. stramonium (De-la-Cruz et al., 2020).

Table 2 Tropane alkaloids identified in oral secretions of Lema daturaphila.

	Alkaloid	Formula	RT (min)	m/z	MS Ref.	
1	3-tigloyloxy-6-hydroxytropane	C13H21NO3	10.1	240.1549	Witte, Müller & Arfmann (1987)	
2	Apoatropine	C17H21NO2	10.4	272.1641	Witte, Müller & Arfmann (1987)	
3	Atropine (Hyoscyamine)	C17H23NO3	8.7	290.1751	Witte, Müller & Arfmann (1987)	
4	Scopolamine	C17H21NO4	15	304.1543	Witte, Müller & Arfmann (1987)	
Note:

We used the combined oral secretions of 10–15 larvae (1–5 µL per larvae) reared in the laboratory and originally collected from the Bernal population in 2019. We performed a unique analysis due to the difficulty of obtaining a bigger sample of oral secretions. RT, the retention time of each alkaloid; m/z, mass/charge; MS, mass spectrometry reference.

Discussion

Our findings demonstrate that parasitoids exert strong selective pressures on the survival of Lema daturaphila populations by attacking both eggs and larval stages. In Central Mexico, the interaction between the herbivore and its parasitoids varies considerably among populations. In some, egg parasitoids are more abundant than larval parasitoids, and vice versa. These differences appear to influence the dominance of L. daturaphila as the main herbivore of D. stramonium and align with findings on spatial variation in the plant-herbivore interaction (Castillo et al., 2013, 2014, 2015). While it is difficult to determine the benefits of parasitoid presence for D. stramonium or whether they represent an effective strategy against L. daturaphila herbivory, these findings provide insights into the interaction and are crucial for identifying potential coevolutionary hotspots.

Parasitoid species

Accurate species identification is crucial for understanding ecological and evolutionary relationships (Smith et al., 2008). Despite being a ubiquitous and diverse group, parasitoids, especially micro-Hymenoptera and Tachinidae, are still under-described (Forbes et al., 2018; Dindo & Nakamura, 2018). Emersonella lemae parasitizes the eggs of chrysomelid beetles in America, particularly in the Neotropics. This is one of the two species of Emersonella reported in the Neartic Region (Alvarenga et al., 2015), primarily in the United States, and represents a new record for Mexico (Noyes, 2019).

Tachinidae flies, the second largest group of parasitoids after Hymenoptera, remain poorly understood in terms of their biology (Stireman, 2002; Dindo & Nakamura, 2018). Genera such as Patelloa and Pseudochaeta are found throughout America, primarily parasitizing Lepidoptera. Records of Patelloa leucaniae attacking Lema exist in the U.S. and Canada (Arnaud, 1978), but it is a new record for L. daturaphila in Mexico (Zetina et al., 2018). The genus Pseudochaeta contains 26 species mainly from the U.S., and both the genus and host are new records for Mexico (Arnaud, 1978; O’Hara & Wood, 2004). Species of the Winthemia genus are widely used as a biological control in economically important crops. In Mexico, there are species such as W. imitator, W. montana, or W. texana (Guimarães, 1972; Zetina et al., 2018), L. daturaphila is a new record as their host. Finally, two species of Heliodorus (H. cochisensis and H. vexillifer) are known from the U.S. (O’Hara & Wood, 2004), and both the genus and the host L. daturaphila represent new records for Mexico.

Previous studies in the Pedregal population reported a tachinid fly parasitizing L. daturaphila larvae (Cabrales-Vargas, 1991). In our study, we identified an undetermined Ichneumonidae wasp. In the Toluca population, we found a hyperparasitoid wasp from the genus Mesochorus, which has a cosmopolitan distribution. This genus belongs to the Mesochorinae family, known for being hyperparasitoids of Braconidae, Tachinidae, and Ichneumonidae (Dasch, 1974). This is the first record of a Mesochorus species parasitizing L. daturaphila parasitoids, making it a new record.

Population variability in parasitoid infestation

Before this study, there had been no reports of egg parasitoids attacking L. daturaphila in Mexican populations of D. stramonium. Emersonella lemae was especially abundant at Tzintzuntzan, Valsequillo, Pedregal, Requena and Texcoco during both years. High host egg availability and kairomone concentration could explain E. lemae’s preference for these sites (van Alphen, Bernstein & Driessen, 2003; Fatouros et al., 2008; Iranipour et al., 2020). Its high incidence corresponded with a scarcity of L. daturaphila larvae and adults, which were not the main herbivores in previous years (Castillo et al., 2014). In addition to low damage, these D. stramonium populations have shown negative selection for direct defenses, which are ineffective against L. daturaphila (Castillo et al., 2014). Since Emersonella lemae is known to be an effective regulator of L. daturaphila in other populations (Girault, 1916; Chittenden, 1924), D. stramonium could attract this parasitoid when direct defenses are no longer effective (Ballhorn et al., 2008).

It is evident that E. lemae also disperses effectively within Mexican L. daturaphila populations. It was particularly successful at higher temperatures and precipitation, but weather conditions could limit its reproductive success at higher altitudes (Weisser, Volkl & Hassell, 1997; Péré, Jactel & Kenis, 2013), as occurs in Toluca and Tlaxiaca populations. This divergent pattern may also reflect the dispersal abilities of the parasitoid wasp, which because of its small size probably moves passively through the wind and could be limited to access at higher altitudes (Fatouros et al., 2008).

Except for Pedregal, Tachinidae flies were found in all studied populations. Tlaxiaca, Toluca, and Teotihuacan were the most parasitized and historically have been dominated by the herbivory of L. daturaphila (Castillo et al., 2014). Tachinidae flies utilize long-range cues to locate their hosts, employing an efficient searching strategy (Hanyu et al., 2009; Dindo & Nakamura, 2018). Even at a low density, they could detect larvae of L. daturaphila in patches where E. lemae kill most eggs. This seems to be the case in populations with high levels of infested egg clutches. Parasitoids usually compete for the host, which serves as a common resource. Although several species could coexist using different development stages of the same host (as occurs with wasps and flies in L. daturaphila), eventually parasitoids that attack a stage will affect the densities of subsequent stages by increasing host mortality (Briggs, Nisbet & Murdoch, 1993). The decline of larval individuals and its negative relationship with the emergence of wasps was clear with statistical analysis, but it also was a continuous observation in the field. Despite the high number of eggs and the frequent presence of adult L. daturaphila in populations such as Valsequillo, Pedregal, and Requena, it was almost impossible to find larvae and consequently, parasitoid flies. These findings underscore the importance of E. lemae in regulating L. daturaphila populations and their impact on interactions established with other parasitoids. In this complex geographic scenario, some populations (i.e., Tzintzuntzan, Valsequillo, or Requena) could function as hotspots, where the interaction with E. lemae shows strong reciprocal selection, while others are dominated by the interaction with parasitoid flies (i.e., Toluca or Tlaxiaca) (Thompson, 2005).

Analysis of oral secretions

In D. stramonium populations, it is usual to observe tachinid flies approaching L. daturaphila’s larvae but avoiding oral secretions from the head. The chemical composition of these excretions can vary with the insect food and is important for all members of a tritrophic system (Alborn et al., 1997; Sword, 2001; Grant, 2006; Zvereva & Kozlov, 2016). They can function as elicitors for the plant and may be essential to differentiate between mechanical and insect damage (Neveu et al., 2002). However, they can also aid in insect digestion, immunity, the suppression of plant defenses (Rivera-Vega et al., 2017) or be a chemical mechanism of anti-predation for herbivores (Pasteels, Braekman & Daloze, 1988; Karban & Agrawal, 2002).

Oral secretions can contain toxic compounds that insects get from food (Calcagno et al., 2004). Some specialist herbivores release this gut content when they are attacked, affecting natural enemies with plant metabolites (Rowell-Rahier & Pasteels, 1992; Grant, 2006). Because trophic specialization of L. daturaphila to D. stramonium and its resistance to many alkaloids (De-la-Cruz et al., 2020), larvae could use the substances present in its host plant to complement the defense behavior they perform before potential attacks by flies (Omer-Cooper & Miles, 1951). While there may be an intriguing relationship between specialization and the defenses of L. daturaphila, it is crucial to experimentally determine which role oral secretions play in the tri-trophic interaction.

Conclusions

This study provides evidence that parasitoids regulate L. daturaphila abundance in central Mexico, thereby influencing its interaction with the host plant D. stramonium and other associated insects. Accurate identification of parasitoids and some aspects of their natural histories were crucial for comprehending their roles in each locality. This spatial variation suggests a geographic mosaic of interactions, with some places dominated by E. lemae and others by Tachinidae flies. These differences contribute to the geographic patterns of D. stramonium herbivory and highlight the need for future studies to evaluate the role of indirect defenses, such as volatile compounds, in attracting parasitoids. While the composition of L. daturaphila’s oral secretions suggest a potential defensive role, further intensive research is necessary.

Supplemental Information

Supplemental Information 1 Parasitoid and hyperparasitoid wasps on Lema daturaphila larvae.

Ichneoumonidae parasitoid found in Pedregal population (a), hyperparasitoid emerging from a pupae cocoon (b), and lateral view of Mesochorus sp. (c).

Supplemental Information 2 Datura stramonium and Lema daturaphila populations sampled in Central Mexico.

Geographic location and average climatic conditions per year of the sampled localities. P = annual precipitation (mm) and T = annual temperature (°C).

Supplemental Information 3 Lema daturaphila individuals collected per population in 2018 and 2019.

Number of egg clutches, eggs, and larvae collected in each locality during both years. The last row shows the data summed per year.

Supplemental Information 4 Variation in the number of eggs per clutch of Lema daturaphila among populations and years.

Estimated values for the number of Lema daturaphila eggs per clutch based on population, year, and their interaction. The estimates were obtained using a negative binomial generalized linear model, with Bernal as the reference population. The model explains 19.23% of the variance, and the interaction between population and year is indicated by an asterisk in the first column.

Supplemental Information 5 Variation in Lema daturaphila egg clutches collected per population in 2018 and 2019.

Mean, median, and standard deviation in the number of eggs per clutch of Lema daturaphila in all sampled populations.

Supplemental Information 6 Variation in the parasitism of Lema daturaphila egg clutches by Emersonella lemae.

Estimated values for the parasitism of Lema daturaphila clutches based on population, year, and their interaction. The estimates were obtained using a binomial generalized linear model, with Bernal as the reference population. The model explains 56.6% of the variance, and the interaction between population and year is indicated by an asterisk in the first column.

Supplemental Information 7 Regression analysis of the probability of parasitoidism by Emersonella lemae based on the clutch size of Lema daturaphila.

Estimates were calculated with a binomial generalized model. In 2019, the clutch size of Lema daturaphila had a significant effect on the probability of being parasitized (GLM, p = 0.0254).

Supplemental Information 8 Population variation in the number of emerged parasitoid wasps of Emersonella lemae and its relationship with clutch size of Lema daturaphila per year.

Estimated values about the number of emerged Emersonella lemae. The number of eggs per clutch and the population were used as predictors. Estimates were calculated with a negative binomial generalized linear model and back-transformed to the original measure scale. For 2018, populations were statistically different from Bernal, the reference population. The model explains 38.05% of the variance for 2018 and 66.87% of the variance for 2019.

Supplemental Information 9 Regression analysis of the relationship between the number of larvae per population and its effect on the number of emerged parasitoid flies each year.

Estimated values about the number of emerged parasitoid flies were calculated with a Poisson generalized linear model and back-transformed to the original measure scale. The model explains 89.71% of the variance for 2018 and 79.02% of the variance for 2019.

We are very grateful to the specialists who helped us with taxonomic determination: Dr. Enrique Ruiz Cancino, Dr. Dulce H. Zetina, for all your help with flies, Dr. Christer Hansson, for your disposition with our specimens, and the E. lemae photos. We thank the members of the Lab of Ecological Genetics and Evolution for their help in the field and Dr. Graciela García for her help. Also, we thank María Berenit Mendoza Garfias for the advice and help with the use of MEB.

Additional Information and Declarations

Competing Interests

The authors declare that they have no competing interests.

Author Contributions

Carol Estefanía Villanueva-Hernández conceived and designed the experiments, performed the experiments, analyzed the data, prepared figures and/or tables, authored or reviewed drafts of the article, and approved the final draft.

Juan Núñez-Farfán conceived and designed the experiments, authored or reviewed drafts of the article, and approved the final draft.

Field Study Permissions

The following information was supplied relating to field study approvals (i.e., approving body and any reference numbers):

Field experiments were approved by the SEMARNAT, DIRECCIÓN GENERAL DE VIDA SILVESTRE Oficio N° GPA/DGVS/011980/17.

Data Availability

The following information was supplied regarding data availability:

The data is available at Figshare: Villanueva, Estefanía; Nuñez-Farfan, Juan (2024). Database Lema daturaphila_CEVH. figshare. Dataset. https://doi.org/10.6084/m9.figshare.22297624.v1.

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
