# Peer review of "Searching for a common host: parasitoids of Lema daturaphila on Datura stramonium in Central Mexico"

_PeerJ, doi:10.7717/peerj.18675_

## Round 0.1 · original submission · Major Revisions

Dear authors,

After receiving the comments of two reviewers, both agree that the manuscript is adequate for the journal and with important contributions, but some major corrections need to be made before it can be accepted. Both reviewers have important observations about the introduction, mainly due to the lack of the hypothesis, the discussion and the conclusions, as well as in the figures and tables.

Best regards,

Armando Sunny.

·

Basic reporting

The authors present the results of research aimed at testing how the presence of parasitoids of an herbivore feeding on Datura stramonium (Lema daturaphila) modifies the arms race between plants and insects. The main premise of this study is the lack of knowledge about the effect of natural enemies of herbivorous insects on
Interactions between a plant and its herbivore.
The authors present results indicating the presence of parasitoids on L. daturaphila eggs and larvae. The authors investigated the existence of the above interaction in 11 populations of D. stramonium and state that they have evidence to conclude that in some populations egg parasitoids dominated over larval parasitoids and vice versa. The authors suggest that this was the result of competition for limited resources and that it may influence competition between parasitoids.
In my opinion, the problem should be presented in a slightly different way, for example: The production of defensive substances protects the plant from attack by most generalist herbivores. However, some herbivores develop adaptations that allow them to find the plant through the scents of the defensive substances it produces, and to neutralize the toxic effects of these substances. Defensive substances can be volatile. Herbivores are not necessarily attracted to the smell of a given defensive substance, but also to another chemical compound that does not perform a defensive function. The herbivore's detection of a specific chemical compound produced by a plant promotes very specific adaptations in that herbivore, which are costly. In accordance with the Second Low of Thermodynamics, the high costs of such adaptations make it less likely to develop adaptations to detect other plants and detoxify their chemical compounds, which makes such an herbivore a specialist. High pressure of generalists and low pressure of specialist are selection factors favouring high concentrations of defensive substances in plant tissues. Low generalist pressure and high specialist pressure are selection factors favouring a low concentration of defensive substances in the plant tissue. The ratio of the pressure intensity of generalists and specialists is a selection pressure that shapes the concentration of the defensive substance and the level of tissue damage. Parasitoids modify this pattern. For example, reducing the pressure of a specialist by increasing his parasitism, with high pressure of generalists will cause an evolutionary response in the form of an increase in the concentration of the defensive substance in tissues and a decrease in the proportion of damaged tissues. Various combinations of generalist, specialist and parasitoid pressure intensities can be studied in variable environments that differ in the rate of primary production and, consequently, in the composition of plant, herbivorous and other organism communities. This phenomenon remains almost unexplored.
The authors do not explain why they conducted the research in different plant populations. One can guess that the reason was to study how frequent cases of herbivorous parasitization by parasitoids are. The authors provide basic geographic and meteorological data of the research sites, but do not address in the text the productivity of these sites, which may be proportional to the average amount of precipitation and average temperature. They also do not discuss differences in the proportion of parasitic broods and larvae, and differences in herbivore egg mortality levels between study sites.
I consider that all the relevant literature was cited by the authors.

Provided the lack of theoretical background referring the use of different study sites, I consider that Figure 6 either is not relevant, or the authors do not know how to present its relevance.

Charts 7 and 10 are descriptive and do not contribute to the understanding of the biological problem presented here. Probably, they will gain importance if the authors explain the idea behind the comparison of different study sites.

Many of the figures and tables are not well labelled: essential information is lacking (I described these shortcomings in the “Experimental design” section).

I am not a native English speaker, but to the best of my knowledge the manuscript is using professional English correctly.

Raw data was supplied by the authors. These data appear to be robust.

Experimental design

The scope of the journal is rather broad, so I consider that the subject matter of the manuscript is consistent with the scope of the journal.

The research question is not sufficiently clear. It does not address existing hypotheses/theories of defence in plants. Many sections of the manuscript are descriptive. Due to conceptual and structural shortcomings and insufficient reference to existing hypotheses of defence in plants against herbivory, the manuscript does not fill existing gaps in this field, although it duly identifies these gaps.

The appropriate model fot the type of data presented in the Figures 8 and 9 is the standardized major axis (SMA) and not the ordinary least squares (OLS), as the authors did not control the independent variable.

The methods do not clearly state whether the data used in the regressions were transformed in any way. From the description of the results, one can guess that it was not. While the fit of the model to the data is not affected by the fact that both variables are discrete, the significance test (I suspect it was ANOVA in regression, because it is not described explicitly) should be carried out on continuous data if no numerical method resistant to discontinuity was used (Falster et al., 2003; Sokal and Rohlf, 1995; Warton et al., 2006).

The quality and the descriptions of the Figures 1 through 5 are OK.

Figure 6, and Figures 8 and 9, should contain information on the statistical significance of the models used. Figure 6 should contain information on what post hoc test was used. Figures 8 and 9 should contain the equations of the regression lines in each figure and information on which regression model was used (standardized major axis (SMA), ordinary least squares (OLS)). The authors do not explain why most of the regression equations were adjusted to the 2019 data and only two, to the 2018 data (Figure 8). Nor do they explain why the equations for different locations were fitted to different numbers of pairs of levels of variables. Figure 10 should include an explanation, either in the text of the manuscript or in the figure legend, why the regression analysis for herbivorous larvae has not been split by study site.

Charts 7 and 10 should contain information specifying that the data are average values per year.

The description of the Table 1 should inform that the data are per year (i.e., average per year precipitation, average per year temperature (or, yearly). Table 2 should contain the information that the data was summed per year.

The description of the Tables 3 and 4 should be different, for example: “The effect of population on the average per clutch egg number in 2018 and 2019” (Table 3: Why the authors did not compare it between years?), and for example “The effect of population and herbivore egg number on the average per season (per year?) number of emerged parasitoid wasps in 2018 and 2019” (Table 4).

Table 5 should contain the information concerning the unit of measurement of the study object (how many larvae, time interval during which analysed larvae were collected, etc.).

The authors performed the investigation in accordance with high technical and ethical standards. Particularly, the authors had a permit from the Ministry of Environment and Natural Resources, Mexico (SEMARNAT), to obtain samples in the field.

The authors carried out the lab and field work in accordance with the existing standards.

Neither the statistical methods the results based upon these methods were described with sufficient details.

Validity of the findings

Since the authors do not refer to existing hypotheses about plant defense against herbivores, the validity of their discovery is diminished.

The text does not directly explain why the authors investigated the relationship between the number of herbivorous eggs or larvae and the number of parasitoids. For both developmental stages, the authors found a positive relationship between these variables, but do not discuss the implications of this finding. If a larger number of larvae or eggs of herbivores attracts a larger number of parasitoids, this is a numerically obvious and not very interesting result. However, the analysis of the regression slopes may indicate the existence of interesting relationships. These slopes are impossible to compute for peerage purposes, as I have not found the coefficients of the regression equations either in the text or in the supplementary material. They are difficult to visualize in the figures given, as the dependent axis is shorter than the independent axis. The slope of the equation 45º means that the number of parasitoids increases proportionally to the number of eggs or larvae. A higher slope indicates a parasitoid’s preference for larger numbers of eggs or larvae, and a slope smaller than 45º, indicates, the existence of a mechanism that reduces parasitoid’s efficiency, e.g. (i) failure of the parasitoid to infect host eggs when eggs are numerous, or, for example, (ii) the existence of a larval defence mechanism when present in large numbers. With one exception (Texcoco 2019), the relationship between the number of parasitoids and the number of eggs was linear, and the deviations from the regression equations did not show a clear trend: due to the lack of information about the slope of the regression equations, it is difficult to interpret these results. However, the relationship between the number of parasitoid flies and the number of L. daturaphila larvae suggests the existence of a trend. The equation for 2018 data can be explained, as proposed by (ii). For 2019, the regression equation seems to fit the data poorly, and perhaps a quadratic or polynomial equation would fit the data better. The results show that low and high numbers of L. daturaphila larvae attract proportionally more parasitoid flies than medium numbers of larvae.

Also, slope/intercept comparison among study sites would help to test for the among-site differences in the intensity of the relationship presented on the Figure 8. I suspect that the authors did not perform such comparison because they did not address this question explicitly (I do not understand why).

In the line 352 the authors wrote: “Thus, the presence of these alkaloids in the oral secretions of L. daturaphila could suggest that this herbivore can take advantage of its host plant defences for its own benefit. This might be the reason behind the larvae behaviour to present oral secretions before potential attacks by enemies such as parasitoids.” It seems that the results show something completely different: as for the high number of herbivore larvae, the number of parasitoid flies is higher than the fitted regression line, this may be due to the attraction of the parasitoid by the larvae, e.g. by the alkaloids secreted by them. This could be demonstrated if a higher concentration of alkaloids attracted more parasitoids. Apart from mentioning that herbivorous larvae accumulated alkaloids, the authors do not use alkaloid data. It should be tested whether a higher density of larvae does not reduce the concentration of alkaloids in the digestive system of the larvae. If not, then the hypothesis of a side effect of alkaloid accumulation by larvae in the form of "attraction" to a specialized parasitoid would be very novel. The problem with this explanation is the higher-than-expected parasitism for low larval population densities. I think this may be due to the fact that it is easier to parasitize the larvae when there are few of them. Being specialist parasitoid involves similar proximate mechanism as being specialist herbivore: specialist parasitoid “learns” how to detect its hosts using their alkaloids and should neutralize their harmful effect.

A result completely unnoticed by the authors is a higher dispersion of residuals for higher larval population densities. This may be consistent with the prediction that parasitism is easier when there are few larvae. For moderate and high larvae density, the number of parasitoid flies can be either lower or higher. For low larvae densities, it was always low. Why?

It is difficult to deduce from the text which results show or suggest competition between the two species of parasitoids. The easiest way to demonstrate this result would be, for example, to show that the prevalence (incidence) of one type of paraitoid is inversely correlated with the incidence of another parasitoid.

I conducted an analysis of the available data, read roughly from the data presented in Figures 7 and 9, and the result does not show the existence of a negative relationship between the percentage of parasitized eggs and larvae for any year:















Data for 2018 showed even a weak positive relationship between the two variables.

This is not straightforward from the text, why the authors evaluated the mortality by parasitoids per egg clutch (Lines 144-145). The authors did not refer this result to any abiotic variable associated to the environmental productivity. For example, at first glance, it seems that higher mortalities occurred when the average (per year?) precipitation interval is between 600 and 700 cubic mm. Raquena in 2019 presented a high mortality, even when the productivity at this site was high. Probably the use of evapotranspiration would better express the productivity of each site. Also, perhaps using a structural equation model would help to better test whether abiotic variables influenced egg mortality (Lefcheck, 2016).

The discussion in the lines 293-340 is vague.
The research question implicitly assumes that greater negative effects of the parasitoid on the herbivorous imply less damage to plant tissues, which may be true but has not been shown in this study. I suspect that the authors' earlier work contains data that could help test this hypothesis.

The “Conclusion” section contains considerable misconceptions. Contrary to what the authors stated, the study is far from demonstrating the existence of coevolution between the host plant, the specialist herbivore and the parasitoids that attack the herbivore. For this, it is necessary to demonstrate that the parasitoid decreases the negative effect of the specialist herbivore on the damage exerted on plant tissue and that the decrease in this damage is reflected in the decrease in the negative effect of herbivory on fitness. The authors have demonstrated the existence of a varied pressure of parasitoids in their area of distribution, but they have not demonstrated that these parasitoids constitute a strong selection pressure. I do not agree that a conclusion can be drawn that the larvae of the specialist herbivore mounted an anti-parasitoid defense: as I described in this section of the review, it is easier to demonstrate that there is evidence derived from this study, which suggests that the alkaloids contained in the larvae's digestive system they are used by the parasitoid to find the larvae of the herbivore. The conclusion: “Finally, the discovery of new interactions in the system Datura-Lema underscores the importance of ecological interactions in the promotion and maintenance of species diversity” goes too far. An alternate scenario is possible: A high efficiency of the parasitoid in reducing the population of the specialist herbivore will promote the decrease of the abundance of the latter species and consequently, promote the concentration of alkaloids in the plant tissue in the absence of this specialist and, therefore, negatively affect the abundance and richness of the generalist herbivores that are going to be eliminated or repelled by these alkaloids.

Additional comments

4. Comments on strengths and weaknesses of the manuscript

The problem taken up by the authors is very interesting and relatively novel, but the authors do not refer the obtained results to any of the theories of plant defence against herbivory. This is a considerable weakness of this manuscript.

I consider that the topic of the manuscript is very important, however, I found a disorder and a lack of a clear theoretical background in the text. Particularly, it lacks a hypothesis-based approach.

The manuscript can be accepted for publication only after considerable changes.

·

Basic reporting

The manuscript Searching for a common host: parisitoids of Lema daturaphila on Datura stramonium in Central Mexico is well written using professional English. It addresses the knowledge of the interaction between L. daturaphila and D. stramonium including the study of the third trophic level, specifically, the parasitoids that participate in this interaction in different locations in central Mexico. This aspect is very important because the spatial variation of tritrophic interactions is rarely studied.
The introduction is well structured, and the background puts the study in context, the references are appropriate. However, the analysis of the oral secretions of L. daturaphila should be better justified mainly because one of the most abundant parasitoids of L. daturaphila’s is an egg parasitoid.
As soon as figures is concern, these are adequate, of good quality, well labeled and described.
Raw data are complete and ordered.

Experimental design

The study is original and within the scope of the journal. Research question well defined and relevant. The question addressed in this research fills the gap regarding the knowledge of the parasitoids of a plant species whose anti-herbivore defenses vary geographically. Methods are sufficiently described, although some details on the rearing of larvae from which oral secretions were obtained are lacking, specifically, what plants were these larvae fed on? This is important because the feeding of the larvae could alter the content of the oral secretions.

Validity of the findings

The discovery and description of several species of parasitoids as well as the data obtained on the behavior and habits of these insects make this study valuable. The methods used, as well as the statistical treatment of the data obtained, are adequate.
Conclusions agree with the results found.

Additional comments

The manuscript is well presented and well written. The conclusions are in accordance with the original study question. The only observation I have regarding this work is about the study of the oral secretions of L. daturaphila larvae. The analysis of the oral secretions of L. daturaphila should be better justified. Some questions regarding this topic are: From which locality were the plants fed the larvae from which the oral secretions were collected?
Would changes in these secretions be expected between localities?
Obtaining these secretions can be a difficult task, however, could they have been studied by locality?
Rivera-Vega et al (2017. PLOS ONE, https://doi.org/10.1371/journal.pone.0182636) have shown that salivary glands of cabbage looper larvae are strongly responsive to diet. It is possible that these oral secretion could be playing a main role in host recognition by parasitoids.
We must also consider that one of the most important parasitoids of L. daturaphila (Emersonella lemae) ia an egg parasitoids, so oral secretions would only be used as a defense against natural enemies in the case of larval parasitoids.
Some questions and typing errors:
Line 78: I suggest removing “up to”, (how can there be damage greater than 100%)
Line 109: Did these leaves come from the same place where the clutches were collected?
Line 165: the presence of some instead of the presence some
Line 274: Ichneumonidae instead of Ichenumonidae

---

## Round 0.2 · Major Revisions

Dear Authors,

Thank you for submitting your revisions. However, one of the reviewers believes that additional major revisions are needed before your manuscript can be accepted for publication in PeerJ. We encourage you to address these concerns, and we look forward to receiving your updated manuscript soon.

Best regards,
Armando Sunny

·

Basic reporting

The current text has been fundamentally changed. Therefore, I conducted the review almost as if it were a new article.

The manuscript examines a very interesting case of interaction between a plant, a herbivore and a parasitoid.

The text mixes information that is important for understanding and explaining the existence of this interaction from the point of view of natural selection, with unimportant information that distracts the reader from the main issue. Some results (i.e. Figs. 7 and 8) do not, in my opinion, add anything important to the analysis, or they do, but the authors do not show it properly.

The text mixes stories about the natural history of herbivore and parasitoid species, with descriptions related to the main topic. Perhaps it would be better to publish these stories in a specialized entomological or botanical journal, or possibly move them to a supporting text.

Experimental design

Please try to use a reductionist approach: the manuscript describes so many different variables that are not analyzed but only mentioned or analyzed but not very important for the most important results, that the reader will probably get lost in the text and have a hard time figuring out what is important, and what is unimportant.

Despite the large amount of research work, the authors clearly do not give importance to the results of chemical composition of insect secretions. These results in the manuscript give the impression of being collateral results, of little importance for explaining the evolution of the studied trophic interaction.

Validity of the findings

Some of the research results are very interesting, others are unnecessarily presented in the manuscript, but almost none of them are adequately discussed. Much of the text is unnecessary. Instead of explaining why some very interesting results took place, the authors cite the results of works that do not serve to explain their results but are rather an unnecessary inserts.

Additional comments

The current version of the manuscript has gained little in terms of clarity of research questions and answers to them. It is still a difficult text to read because of the confusion of more and less important issues.

Instead of trying to explain the obtained results as precisely as the existing data allow, the authors cite the results of other studies that often have little in common with their research. The potential of this very interesting and important research is lost in the imprecise interpretation of the results, poorly written discussion, and the lack of real conclusions.

The authors should separate and perhaps eliminate too descriptive parts of the text and move them to supporting information, precisely define what variables are important to explain the existing interaction and why, and describe what the research plan was. They should also precisely describe the predictions of the outcome of this interaction and try to explain how their results relate to these predictions. In the previous review, I gave an example of such predictions. If these predictions are not exactly those that might be proposed by the authors, the text in question may help in formulating them.

Comments:

Fig. 5. If asterisks denote significant differences between years, this also means that the lack of asterisks denotes the lack of differences. So, the letter “a” is not necessary to denote such a lack of differences.

It is not clear in the Table S3 and Table S5, what the asterisk before 2019 means?

Fig. 6. It is difficult to understand what this figure shows.

Also, the description “Variation in the probability of parasitized egg clutches of Lema daturaphila by Emersonella lemae among populations and years.
Estimated values about the probability of parasitized Lema daturaphila clutches based on the population, year, and their interaction. The estimates were made with a binomial generalized linear model and Bernal was used as the reference population. The interaction was not statistically significant but the model explains 56.6% of the variance.” is confusing and thus, should be more accurate.

How did you obtain the regression line and its confidence intervals?

What the expression “Variation in the probability of parasitized egg clutches of Lema daturaphila by Emersonella lemae among populations and years” means?

If I understood correctly, the positive regression line was attributable mainly to the high probability of parasitism in Tzintzuntzan: the corresponding probability of parasitism is high for large egg number per clutch, with almost no low probability data for such large clutches. Will this regression be still significant with the exclusion of this population?
If I'm right, why isn't this in the discussion?

Fig. 6: Why did you present results only for 2019?

Axes’ descriptions: it would be better to describe it more accurately: “Probability of parasitism by E. lemae” and “Eggs per clutch of L. daturaphila” (or, Egss of L. daturaphila per clutch), or analogous.

Fig. 7. This sentence is very strange: “Bars indicate the number of egg clutches parasitized and non-parasitized byEmersonella lemaeand above each bar, the percentages of parasitized or not clutches corresponding to the total number ofLema daturaphilaegg clutches collected in each population.”
Besides, some words are stuck together. Also, in the first sentences of the Fig. 7 legend.
The same problem occurred in the Fig. 8.

It is difficult to guess what implications Fig. 7 and Fig. 8 have for the rest of the argument. If they have any implications, in my opinion they are not sufficiently reflected in the discussion. I have the impression that without the results presented in these figures the text will become more readable, because these results obscure the most important outcome of this manuscript.

Similar comment concerning Fig. 10.

Lines 380 through 401: It's hard to understand what this text is for?

Lines 403 through 415: This part of the text is unclear.

Lines 416 through 427: After these statements: “In all cases, high levels of infested egg clutches resulted in low amounts of larvae 417 individuals.”, and “The decline of larval
individuals and its negative relationship with the emergence of wasps was clear with the PCA analysis, but it also was a continuous observation in the field.”, ….

…please, try to explain this result instead of discussing why certain populations are more important than others in maintaining the L. daturaphila population: if I understand correctly, explaining this aspect was not the purpose of this study.

Lines 428 through 439: Similar comment.

Lines 440 through 450: The Authors wrote: “In all sampled populations, both eggs and larvae were most highly parasitized at lower or higher abundances. In the former case, clutches with more than 20 eggs were more parasitized, especially during 2019.”

Interesting observation but what is the conclusion?

The Conclusions section does not contain any important conclusions.

The article may be accepted for publication after making corrections, in particular using more precise language, defining and explaining the use of the studied variables, referring to existing hypotheses (theories), separating the descriptive part from the quantitative part, avoiding citing works that do not explain the results, and writing precisely the discussion.

---

## Round 0.3 · accepted · Accept

Dear [Author's Name],

I am pleased to inform you that, following the revisions made in response to the reviewers' comments, your manuscript titled "Searching for a common host: Parasitoids of Lema daturaphila on Datura stramonium in Central Mexico " has been accepted for publication in PeerJ.

The thoughtful and comprehensive revisions you have provided significantly enhanced the quality of the manuscript, addressing all concerns raised during the review process. We appreciate your efforts in making these improvements.

Congratulations on this achievement, and we look forward to seeing your work published soon.

Best regards,
Dr. Armando Sunny